# Single-cell RNA sequencing analysis of human Alzheimer's disease brain samples reveals neuronal and glial specific cells differential expression

**Lilach Soreq**[1]*, **Hannah Bird**[1], **Wael Mohamed**[2,3], **John Hardy**[1,4]

1 Department of Molecular Neuroscience, Institute of Neurology, University College London, London, United Kingdom, 2 Basic Medical Science Department, Kulliyyah of Medicine, International Islamic University Malaysia, Kuantan, Pahang, Malaysia, 3 Clinical Pharmacology Department, Menoufia Medical School, Menoufia University, Menoufia, Egypt, 4 Reta Lila Weston Institute of Neurological Studies, UCL ION, London, United Kingdom

* l.soreq@ucl.ac.uk

**Data Availability Statement:** All relevant data are within the paper and Supporting Information files. Additional data is available here Gene Expression Omnibus (GEO, accession number GSE175814),

## Abstract

Alzheimer's disease is the most common neurological disease worldwide. Unfortunately, there are currently no effective treatment methods nor early detection methods. Furthermore, the disease underlying molecular mechanisms are poorly understood. Global bulk gene expression profiling suggested that the disease is governed by diverse transcriptional regulatory networks. Thus, to identify distinct transcriptional networks impacted into distinct neuronal populations in Alzheimer, we surveyed gene expression differences in over 25,000 single-nuclei collected from the brains of two Alzheimer's in disease patients in Braak stage I and II and age- and gender-matched controls hippocampal brain samples. APOE status was not measured for this study samples (as well as CERAD and THAL scores). Our bioinformatic analysis identified discrete glial, immune, neuronal and vascular cell populations spanning Alzheimer's disease and controls. Astrocytes and microglia displayed the greatest transcriptomic impacts, with the induction of both shared and distinct gene programs.

## Introduction

Alzheimer's disease is the most prevalent neurodegenerative disease worldwide, with increasing number of new cases each year. It is estimated that there are currently 44 million people worldwide living with dementia, disproportionately affecting women. The disease is the currently the 6[th] leading cause of death in the united states, highlighting the essential need for a therapeutic treatment to slow or eradicate disease progression. This disease is molecularly characterized by biochemical lesions of amyloid β plaques and Tau tangles. Additionally, rare mutations in amyloid β protein, amyloid precursor protein, presenilin 1 and 2, *ADAM10* and *ADAM17* have been shown to triggers amyloid-beta accumulation and are sufficient to induce full biochemical and morphological signatures of Alzheimer's disease.

link https://www.ncbi.nlm.nih.gov/geo/query/acc.cgi?acc=GSE175814 release date May 30, 2024. DOI: 10.13140/RG.2.2.22676.01921.

**Funding:** Dr. Lilach Soreq is funded by RoseTrees UK (Stonygate). The project was also partially funded by Alzheimer's Society UK. J.H thanks the Dolby Foundation for support. The funders had no role in study design, data collection and analysis, decision to publish, or preparation of the manuscript.

**Competing interests:** The authors declare no conflict of interest/no competing interests.

**Abbreviations:** AC, Astrocytes; A- β, Amyloid-β; AC, Astrocytes; AD, Alzheimer's Disease; APP, Amyloid Precursor Protein; Braak, Alzheimer's clinical staging; FDR, False Discovery Rate; GEO, Gene Expression Omnibus; GTF, General Transfer Format; GWAS, Genome Wide Association Studies; HCL, Hierarchical clustering/classification; OLGs, Oligodendrocytes; MG, Microglia; MTA, Material Transfer Agreement; NABEC, North American Brain Bank; NC, Neurological controls; NFKB, NF Kappa B; RNA, RiboNucleic Acid; RNA-Seq, RNA Sequencing; scRNA-Seq, single-cell RNA-Sequencing; TF, Transcription Factor; UKBEC, UK Brain Bank Consortium; UMAP, Uniformed Mannford Approximation.

In addition the role of amyloid β, diverse intracellular processes and neuronal cell-types are likely required for Alzheimer's progression. Non-neuronal genes (e.g microglia, MG markers) have been shown to have a central role in Alzheimer's pathology, as exemplified by genome-wide association studies (GWAS) with associated variants having largely glial cell restricted expression [1]. Recent analysis of single-cell RNA-Seq (scRNA-Seq) datasets from the human brain, further highlight an association between microglia and astrocyte pathway perturbations. While mouse models of Alzheimer's and in particular microglia further implicate these cells in Alzheimer's disease-associated pathways, one third of putative Alzheimer's risk genes lack adequate mouse orthologs [2]. Additionally, oligomeric amyloid-β induces a cellular response in human versus MG depleted mice brains [3]. In this study, microglial replacement/repopulation improved cognition in the mice. High expression of high expression of both TMEM119 and P2RY12 was detected. The repopulation did not respond to neuroinflammatory signal. Only a single differentially expressed gene (Cx3Cr1) was detected upon such treatment. Also, Microglia repopulation restores expression of genes associated with cytoskeletal remodelling and neural processes. Most of the downregulated genes in aging that were subsequently restored back to young control levels with microglial replacement are associated with actin cytoskeleton remodelling (Map1a, Map1b, Prepl, Ttll7, Clasp2, and Nfasc) or processes involved in neuronal and synaptic function—including endocytosis (Cltc) and microtubule transport (Kif1a, Kif1b, Kif21a, Dync1h1, and Dnm3)—all important for synaptic vesicle release. We did not found these genes are altered in our data through network analysis. Microglia also regulates neurogenesis. Their elimination and repopulation in aged mice alter dendritic spine densities and neuronal complexities. Their repopulation in aged mice rescues deficits in long-term potentiation.

It is becoming increasingly clear that signals from the central nervous system are required to sustain microglial cellular specification. Notably, loss of those cues dramatically disrupts the microglia phenotype driving them toward an activated cellular state. In addition, some well-known disease-associated genes have been shown to play an essential role in the cross-talk between microglia and other brain cell types (for example, triggering receptor expressed on myeloid cells 2 (e.g *TREM2*)–membrane phospholipids–apolipoprotein E and *CD33*–sialic acid.

We recently reported broad changes in glial gene expression and neuronal and oligodendrocytes cell quantitates in brains of old compared to young individuals [4]. This study included an analysis of 1,231 post mortem brain samples from 134 individuals (ages 16 to 104), profiled by exon microarrays (10 brain regions) [4] and 480 samples from the north American brain bank (NABEC, frontal cortex and cerebellum). We also analysed microglia depleted mice (including comparison of young vs old mice) by analysis of conventional microarrays, highlighting broad gene expression differences [3]. Additionally, previously disease-associated astrocyte-gene programs were also found as altered in Alzheimer's and aging using single-cell RNA-Seq by other studies, suggesting a linage of astrocytes to genetic and age-related disease factors [5].

In our current study, we have meticulously bioinformatically analysed single cell sequencing data produced from post-mortem brain samples from 2 AD and 2 control brains. Our goal was to detect expression changes in cell- specific marker genes, in the samples of patients compared to control samples (similarly to as done it [6]). Surprisingly, computational analysis of the data showed both shared and cell-type specific expression changes in Alzheimer's genetic programs among additional changes in diverse glial subsets and in particular microglia [7]. The associated transcriptional networks highlight diverse cell-cell signalling induced networks associated with well-established and new transcriptional regulators.

## Materials and methods

A total of 2 Alzheimer's post mortem brain samples and 2 age- and gender- matched control brain samples were obtained from the UK brain bank. The samples were obtained from from anterior hippocampal cortex (HIPP). The patient were in Braak stage I and II for the AD samples. CERAD neuritic plaque score and Thal amyloid phase were not measured thus not reported (also for the control cases) as well as APOE status. The control samples were all free from neurodegenerative brain markers. The age and gender of the samples are as follows: AD, 69 (male), 63 (male) Control, 63 (male), 74 (male). The samples were aged-matched. We are aware that gender may have specific effects on RNA expression however due to cohort limitation this was our choice and we believe that the disease effect on expression changes in larger. Cell Ranger software was used to map the sequenced data (fastq files) to Ensembl database. The ethics approval (MTA) is given in the supplementary material. Additionally, the samples info is given as well as the organized differentials (ST2). Subsequently, single nuclei were isolated from the frozen samples using previously published protocols [8], and we prepared single nucleus RNA-seq libraries using a 10X Genomics Chromium instrument and 10X Genomics Single cell 3' RNA kits (version 3.1) following the manufacturer's recommendations. We sequenced libraries by paired-end sequencing on a NovaSeq 6000 system according to the manufacturer's instructions (Illumina), and at least 200 GB of raw data were obtained per library–using Fastq generated (for GEO upload) using Illumina bcl Convert v3.7.5. Fastq files were processed using 10X Genomics CellRanger version: reads were aligned to GRCh38 reference human genome and counted using a GTF file including both exon and introns.

Statistical analyses: we included in our current analysis only genes with count value > 10 per gene in each sample. The percentage of reads passing filter was slightly low (68%). Overall output was a bit lower than optimal. There was still over 1.6Bn clusters generated though, which is within the specification for an S1 flow-cell, and base call quality (quality control) was excellent: 97% >Q30. The read counts mapping to genome occupancy rate was also fairy low (63–81%).The number of barcodes is as follows: sample 1 (AD, Braak I-II) - 6,223 for sample 2 (control) 6,262 sample 3 (AD Braak I-II) 5,332 for sample 4 (AD, Braak I-II). No other neurodegenerative conditions were found among the AD samples (e.g. CJD/cerebrovascular diseases/other tauopathies).

## Ethical statement

All the Alzheimer's and control samples have approved signed MTA (number T61.16 AND TR73.16) from the university of Edinburgh (e.g curtesy of Mrs. Chris Ann McKenzie and Prof. Colin Smith) dated 25/11/2016 (given under supplementary information). This specific study was reviewed and approved by an institutional review board (ethics committee) before the initiation of the study.

## Brain samples

We have analyzed a total of 4 samples of 2 control (1. Sample ID SD034/15, BBN001.26308 paraffin fixed (Hippocampus) and BBN_24219 (sample ID SD042/14, paraffin fixed, age, 69 sex m PMI 49, and age 63, male PMI 76)–with no clinical abnormalities. The 2 early stage AD patient brain samples brain bank serial numbers are BBN_19604 (sample ID SD037/13) and BBN_1959 age 52 (age 52 PMI 52 anterior HIPP Braak stage I/II, clinical information: 1 parenchymal haemorrhage > 10 mm diameter, lacunar infarct/s, sporadic CAA) and age 77 (anterior HIPP sample ID SD043/13, clinical information: light intracerebral haematoma, small vessel disease large vessel atherosclerosis AD Braak stage II. The AD samples had

Haemorrhage, cerebral amyloid angiopathy and the cases were sporadic with vascular disease. See S1 Table.

**Statistical analyses.**   The iCell8 software was used to generate feature-barcode matrices. We then used R studio and Matlab R21A (including the bioinformatics toolbox) to analyse the data. The number of read counts in the sequenced libraries were 1,718,425, 12,372,392, 13,206,940 and 3,351,685. There were 6,223 barcodes for sample 1, 6,262 for sample 2 3,538 for sample 3 5,332 barcodes and for sample 4, 10,952. Number of features: A1-A4 33,538 each.

*Cluster Analysis* Using AltAnalyze and Cell Harmony, we have analysed the 4 AD and control samples. The Alzheimer's and control brain samples were jointly analysed using the software ICGS2 (AltAnalyze version 2.1.4) with default options, with no cell cycle removal and 5000 PageRank/Louvain down-sampled cells from input sparse matrix files saved to the same input directory (AltAnalyze.py—runICGS yes—species Hs—platform RNASeq—expdir $path–exclude CellCycle no–down-sample 5000).

This analysis identified 30 transcriptionally distinct cell populations (unique marker genes), with cell-type annotation automatically assigned by ICGS2's BioMarker gene-set enrichment database. To better define gene expression differences between AD and control brain samples, cell populations were aggregated based on the assigned cell-identity and similarity of the UMAP coordinate embedding of the clusters (Astrocyte = c25+c23, Inhibitory_CXCL14 = c46 +c32). All control and AD cells were re-classified based on the Marker Finder top-60 markers per cluster centroids of these aggregate clusters, with differential gene expression computed between all AD and control cells, using the software cell Harmony (correlation CutOff rho = 0.5, fold change >1.2, p<0.05, FDR corrected). Our graphical abstract shows the general experimental setup.

## Results

To explore cell-type specific Alzheimer's impacts, we profiled frontal cortex brain region from two age-matched Alzheimer's samples and two controls by single-cell Chromium droplet sequencing (10x Genomics). (note: we also profiled BA41/42,BA 6/8 and anterior hippocampal cortex but this data was not included in the current manuscript to reduce brain-region derived data statistical variability). To simplify our results, we focused our analyses solely on the frontal cortex. Combined unsupervised classification analysis of these samples revealed 28 transcriptionally distinct cell populations from the software ICGS2, with labels corresponding to previous neuronal subsets based on gene-set enrichment analysis. As several of these populations were predominantly associated with a single-sample, we re-ran this analysis to restrict the diversity of clusters (decreased maximum resolution), which produced bar plots of transcriptionally distinct cell type populations spanning the Alzheimer's and control samples. Bar-plots of inhibitory neurons, microglia macroglia/oligodendrocytes astrocytes and inhibitory and excitatory neurons and pericytes are given (blue: control red AD–cell cluster analysis (Fig 1A and 1B). Hierarchical classification on cell specific gene markers classified the altered genes (Fig 1C). We also verified the cell population gene marker specific identities using well-established neuronal cell-population markers (Fig 1B).

Comparing observed cell frequencies within these samples, we generally observed consistent or a high-degree of variability between populations, except in the case of Microglia which were consistently up-regulated between the AD disease and controls (Fig 2 Fisher Exact test p = 5.0E-126 (based on cell Harmony analysis results).

In total, we have found a total of 3,693 differentially expressed genes from all pairwise Alzheimer's cell-type comparisons. Gene upregulation was a general hallmark of Alzheimer's expression impacts, with over 1,700 upregulated genes in astrocytes as compared to only 14

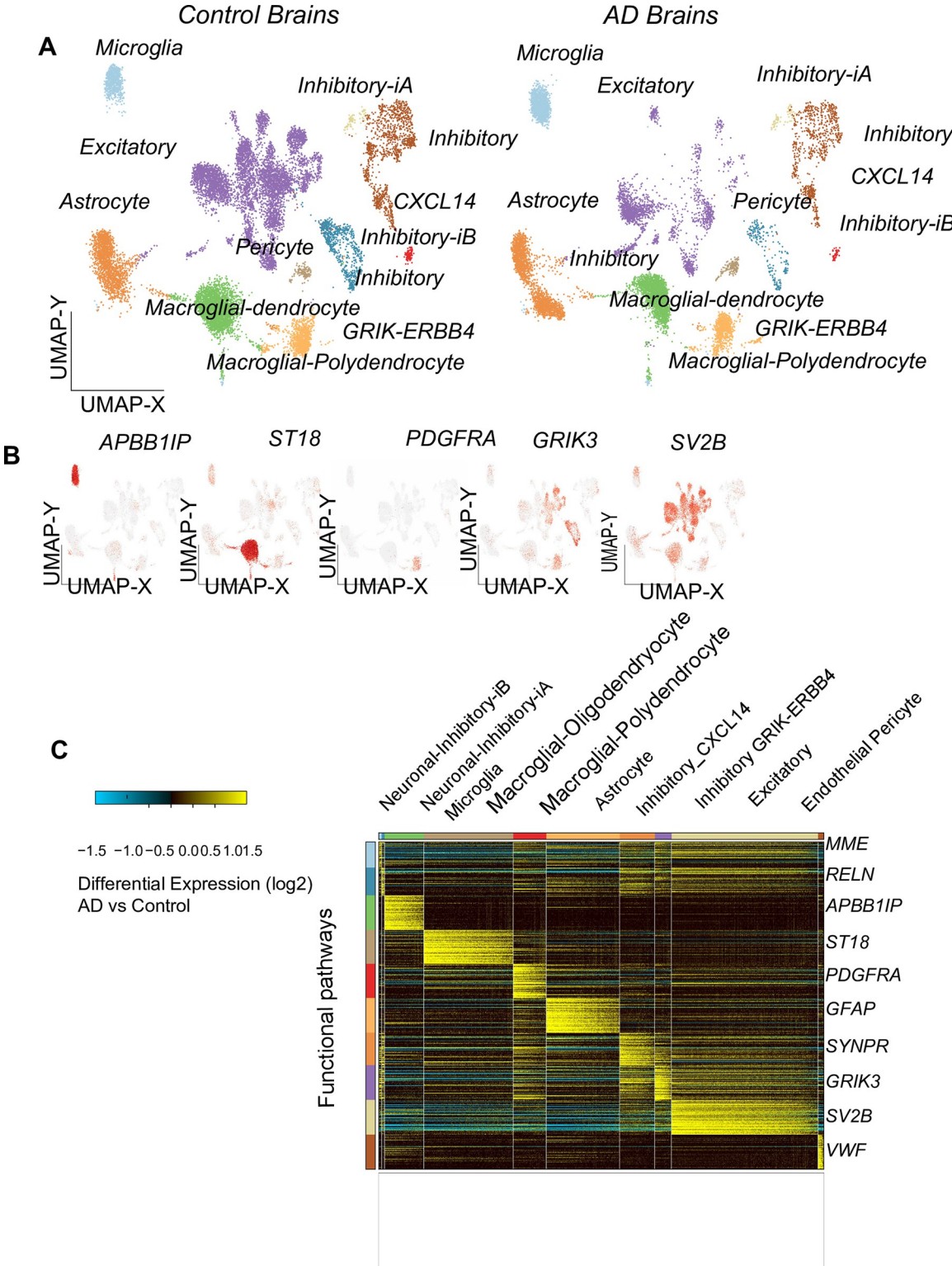

**Fig 1. AltAnalyze analysis of the AD vs control samples.** Results of AltAnalyze (www.altanlyze.org) Python software analysis on the Alzheimer's vs. control RNA-Seq single cell dataset. A) plots of control brain samples (n = 2 each, x axis: UMAP-X, y axis: UMAP-Y. B) UMAP of AD samples highlighting microglia, astrocyte excitatory neurons gene marker clusters.

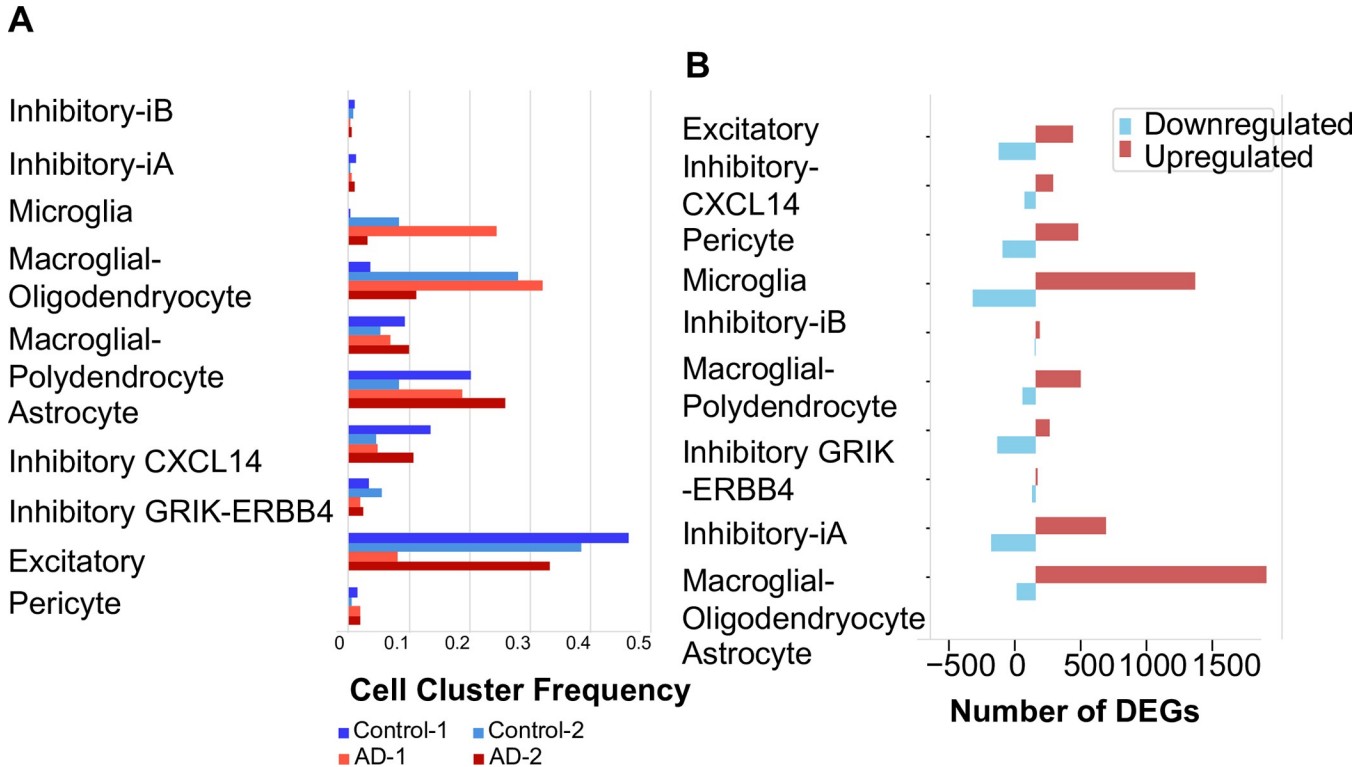

**Fig 2. UMAP plots of 6 genes found as differentially expressed.** A) UMAP plots 6 genes found as altered in the AD brain samples compared to controls (namely *APBB1IP*, *ST18*, *PDGFRA*, *GRIK3*, *SV2B*). B) Hierarchical clustering (HCL) of the 4 AD vs control samples (color bar: Log fold changes, from light blue to yellow). Top line; cell type specific classes, y axis: The clustered differentially expressed gene names. Left: Relevant gene functional categories are noted. Right: Altered gene symbols. Down: Alzheimer's and control brain samples.

upregulated in stromal cells (expression fold change > 1.2, empirical Bayes moderated t-test p<0.05, FDR corrected). Cell-Harmony organized gene expression differences analysis identified common up- and down-regulated genes in Microglia, Macroglial-dendrocytes, Macroglial-Polydendrocytes and Astrocyte cell populations in Alzheimer's samples versus control. Commonly upregulated genes in these cell types include those mediating *PAR1*-mediated thrombin signalling events, cell-projection, axon-guidance, cell-differentiation, Rho protein signal transduction, ubiquitin-dependent protein catabolic process and mitochondrial electron transport, *Beta1 integrin cell surface interactions*, among other processes (GO-Elite, Fisher Exact p<0.05, FDR corrected) (S1 Table). Genes downregulated in these cell include those enriched in transmission across chemical synapses, ion channel binding, neuronal projection and RNA splicing among others.

We generated scatter plots of cell type specific marker genes in Alzheimer's vs control samples (Fig 2A and 2B), and of ordered neuronal cells. We then generated another figure plot of cell type marker genes (Fig 3A), bar plots of different types of neurons (Fig 3B) and another HCL (Fig 3C). We then generated a network of the deregulated genes (Fig 4). We then generated a network of Alzheimer's disease altered genes (Fig 5). These included *NOTCH2* and *MAPT* genes. Red: up regulated Alzheimer's vs control, blue down arrows: red–verified transcription factor targets, grey protein-protein interaction.

While common glial induced transcripts are frequent in AD, cell-specific impacts were most common, with Astrocyte, Microglia, Oligodendrocytes, Excitatory and Pericyte, most frequent, respectively. Notably, astrocytes gene expression changes were most prominent

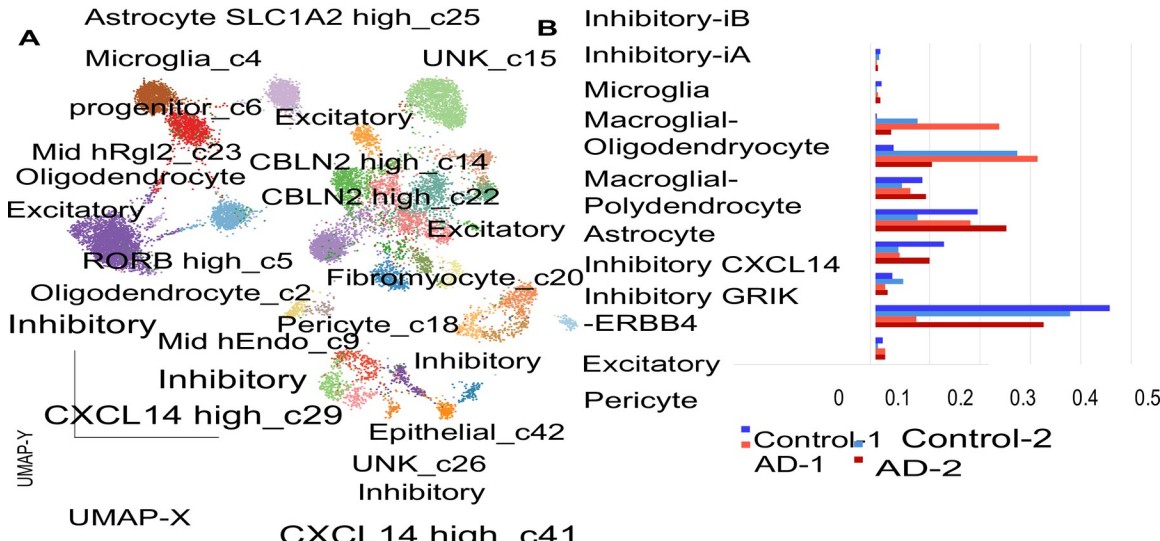

**Fig 3. HCL of the AD and control samples.** A) HCL of the Alzheimer's and control samples (yellow—up regulation, blue/black—down regulation of expression). Top colormap: Range from -1.4 to 1.4 log fold change. Left colormap marks the different neuronal cell groups. Right: The gene symbols and p-values, bottom grey colorbar: Grouping of the samples (white, grey scale and black). P values are indicated on the right. Left: Molecular pathway names and p values top panel: Cell type marker genes right: Altered genes + p values. X axis: AD vs control fold change (from -0.8–0.8).

among these, with pathways associated with diverse extracellular signalling pathways most enriched among upregulated genes.

To identify possible gene regulatory network underlying transcriptional regulation in these distinct sub-cellular populations, we have further explored prior evidenced transcription factor targets relationships from our cell Harmony analysis. Importantly, we observe both common and cell-type specific core transcription factors regulators, with both enriched downstream targets and regulated transcription factors. Upregulated gene networks including of Astrocyte marker genes were found as associated with broad regulation by *HIF1A*, *SOX2*, *NRF1*, and *RB1* genes, respectively (Fig 2). Additionally, microglial marker genes showed higher expression in AD compared to controls. *HIF1A* gene is also a core node in the Microglia and Pericyte gene regulatory work (GRN), whereas *RB1* and was again a core node in Microglia. Beyond these common members, we uniquely find evidence for *ESR1*, *FOS* and *BACH1* regulation in Microglia; *CREB5*, *TCF4* and *MITF* in the first Oligodendrocyte subset; *BACH1* in the second Oligodendrocyte subset; and *STAT3* in Pericytes.

Differential expression analysis of the AD compared to control HIPP single cell expression data revealed differential expression changes in the following cell type specific marker genes: stromal, microglia, oligodendrocytes, astrocytes, inhibitory excitatory and cerebellum. P-valued were calculated for 3,694 genes. Additionally, fold change was calculated per each gene. The organized differentials of specific neuronal cell types were also calculated.

## Comparison to external AD single cell RNA-Seq datasets

To further enlarge the statistical significance of our findings, we have also compared our results to an additional published RNA-Seq data from aging mice brain samples (dataset accession number GSE129788 [9], publicly available for download from the SRA repository (accession number DRP19249, and bio-project PRJNA532831).

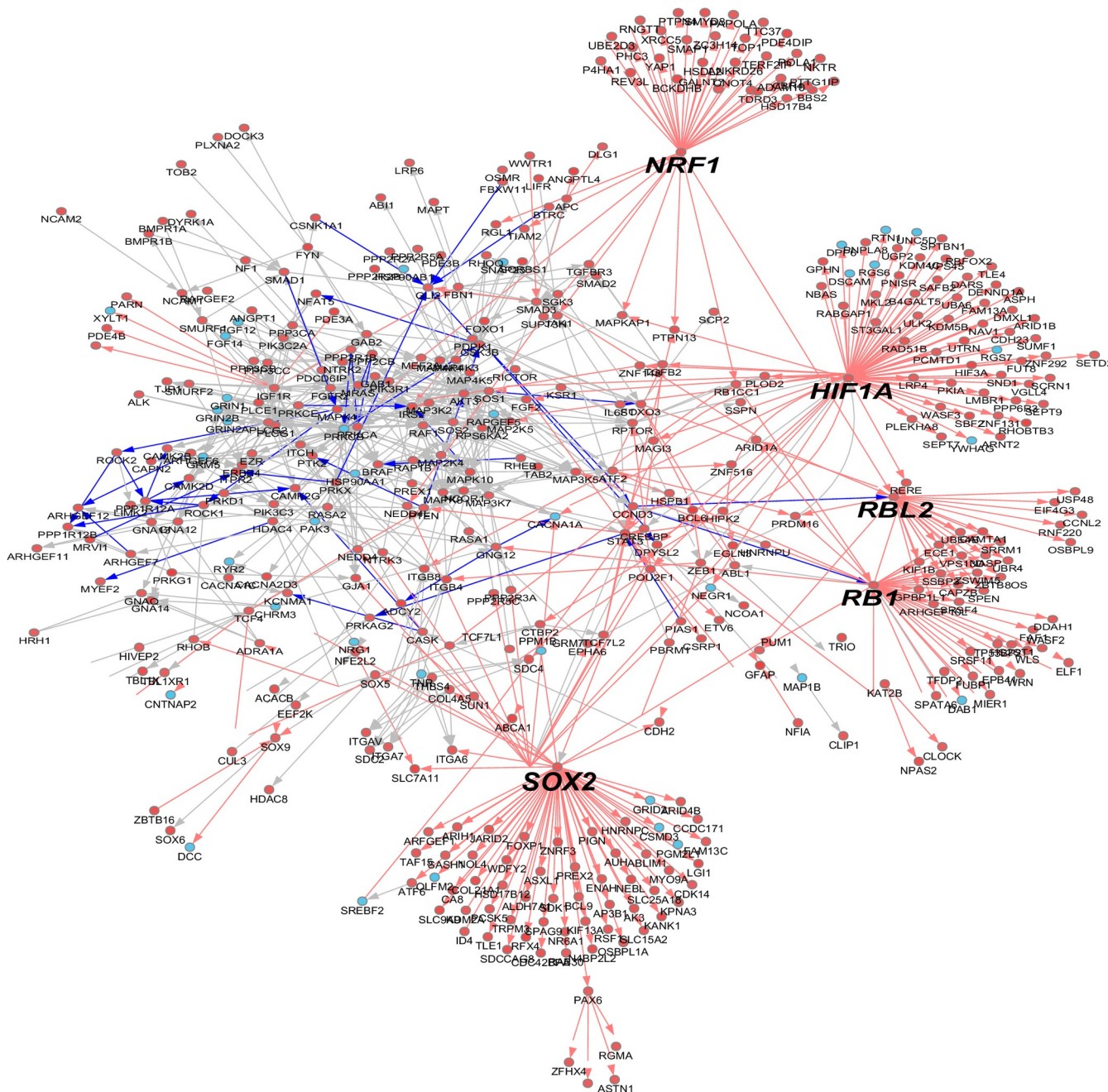

**Fig 4. Gene network.** A) gene network of genes found as differentially expressed between AD to control samples. The most highly abundant macroglia and oligodendrocyte gene markers gene names are shown. For example: *SOX2,NRF1, HIF1A RBL2, RB1*. Blue: Down regulation, red up regulation. Each node corresponds to a gene. The edges: Blue–down regulated gene, red marks up regulated genes. Top column names are marked with the gene groups marker genes names. Distance method: Euclidian. The analysis was done in AltAnalyze program. B) Pericyte gene network. Noted are, *PBX1 HIF1A* and *STAT3*.

A total of overall 16 mice brain samples with raw data for 50,212 single cells and the processed data—409.8 Mb for 37,089 single cells from 16 mice brains was profiled. The platform was Illumina NextSeq 500. These additional analyses revealed analysis identified gene signatures that vary in a coordinated manner across cell types and gene sets that are regulated in a cell-type specific manner, even at times in opposite directions. The analyses of this dataset revealed that aging, rather than inducing a universal program, drives a distinct transcriptional

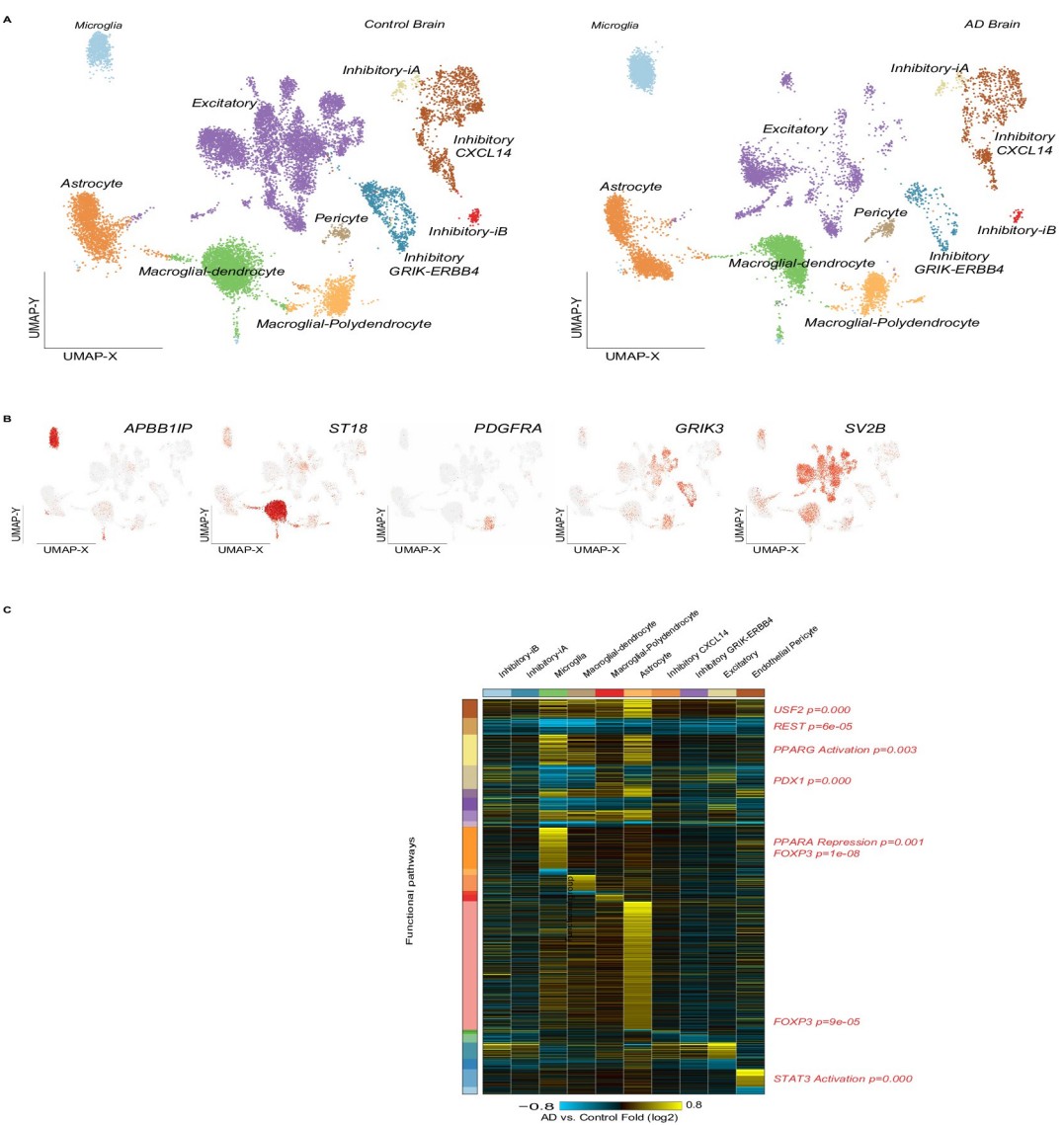

**Fig 5. Microglia gene network.** Gene network A) microglia. Blue: Down regulation in control samples red: Up regulation marked in red arrows–verified transcription factor (TF) targets. Grey arrow: Protein-protein interaction (PPI). Selected gene symbols are marked by larger font size. Example for such genes: *HIF1A*, *PPARA*, *RB1* and *RUNX1*.

course in each cell population, and they highlight key molecular processes, including ribosome biogenesis, underlying brain aging. Specifically, we applied t-test to compare between the disease model and control mice including application of the FDR correction.

Overall, these additional analysed large-scale datasets provide a resource for the neuroscience community that will facilitate additional discoveries directed towards understanding and modifying the aging process. Also, immune lineage changes were detected. these large-scale datasets (accessible online at https://portals.broadinstitute.org/single_cell/study/aging-mouse-brain) provide a resource for the neuroscience community that will facilitate additional discoveries directed towards understanding and modifying the aging process. *GABA_1*, *GLUT_1* neurotransmitter-expressing neuronal subtypes were also analysed. Changes in intracellular communication were also found and in expression of ribosomal genes. Also cell type specific expression patterns were found.

A second dataset for comparison was GEO GSE157827. This data was composed of single-nucleus RNA-sequencing of prefrontal cortex from AD patients and matched healthy controls [10]. We have performed transcriptome analysis of 2 prefrontal cortex tissue samples from AD patients (n = 12) and Neurological control subjects (n = 9). They sampled were composed of 169,496 nuclei: 90,713 and 78,783 nuclei from AD and neurological control brain samples, respectively and clustered the data. Among their findings, these six major cell types expressed the following unique signature genes, which can serve as novel cell-type markers: *ADGRV1*, *GPC5*, and *RYR3* were expressed by astrocytes; *ABCB1* and *EBF1* by endothelial cells; *CBLN2* and *LDB2* by excitatory neurons; *LHFPL3* and *PCDH15* by inhibitory neurons; *LRMDA* and *DOCK8* by microglia; and *PLP1* and *ST18* by oligodendrocytes. Additionally, the proportion of endothelial cells was higher in the Alzheimer's disease samples as compared with neurological control samples. results reveal that the cell type-specific transcriptomic changes in Alzheimer's disease are associated with 4 molecular pathways, specifically: angiogenesis in endothelial cells, immune response in endothelial cells and microglia, myelination in oligodendrocytes, and synaptic signalling in astrocytes and neurons. We have also compared the results with microarray data from large cohort studies that examined samples from the prefrontal cortex (Alzheimer's disease: n = 310; Neurological controls: n = 157) or temporal cortex–TCTX (Alzheimer's disease: n = 106; Neurological controls: n = 135). Among the DEGs identified in our single-cell RNA-seq analysis, 1,113 and 764 genes were significantly differentially expressed in the microarray data from the prefrontal cortex and temporal cortex, respectively. subcluster analysis of microglia identified 13 cell subpopulations; *VGF* signalling was also found as changed. To summarize, the results of our current single-nucleus transcriptomic profiling of Alzheimer's brains are a useful resource for understanding the cellular dysregulation along Alzheimer's progression. In comparison to our data analyses, cell specific changes were commonly found as altered in both studies.

Additionally, we have analysed a 3rd external GEO dataset GSE157827 (composed of a total of 21 samples, 13 Alzheimer's and 9 neurological controls) (Fig 6). The log fold change and p value of cell marker specific genes were plotted. The proportion of astrocytes, oligodendrocytes were plotted. The bottom plots in the figure shows p values and fold change of astrocytes, endothelial cells, excitatory neurons inhibitory neurons oligodendrocytes and microglia gene markers.

## Discussion

Identification of genes involved in Alzheimer's disease is still a major challenge. Cellular heterogenies is still large in Alzheimer. Synaptic signalling is impaired in the disease but the exact underlying molecular networks are largely unclear. Single-cell sequencing present a great advance towards our understanding of human diseases. Here, we sequenced samples from Alzheimer's and control brain samples and analysed the produced single-cell data. We found changes in oligodendrocytes, microglia and other brain cell types. We also detected significantly changed genes including cell specific markers. We will extend our studies in the future to study the immune response in Alzheimer's disease. Our transcriptome profiling revealed changes in endothelial cells, astrocytes and inhibitory neurons. We will also compared our data to Alzheimer's model mice RNA-Seq datasets. By analysis of publicly available Alzheimer's/neurological control datasets have found significant expression changes in cell specific marker genes (e.g. neurons microglia oligodendrocytes inhibitory and excitatory neurons).

Specifically, in this small-case study finds we identified diverse strongly-evidenced gene regulatory networks associated with extracellular induced signalling networks (proteoglycan, *HGF*, *IGF*, *VEGF*, *PAR1*, Estrogen receptor 1 (*ESR1*)), found downstream of *HIF-1*, *NRF1* and

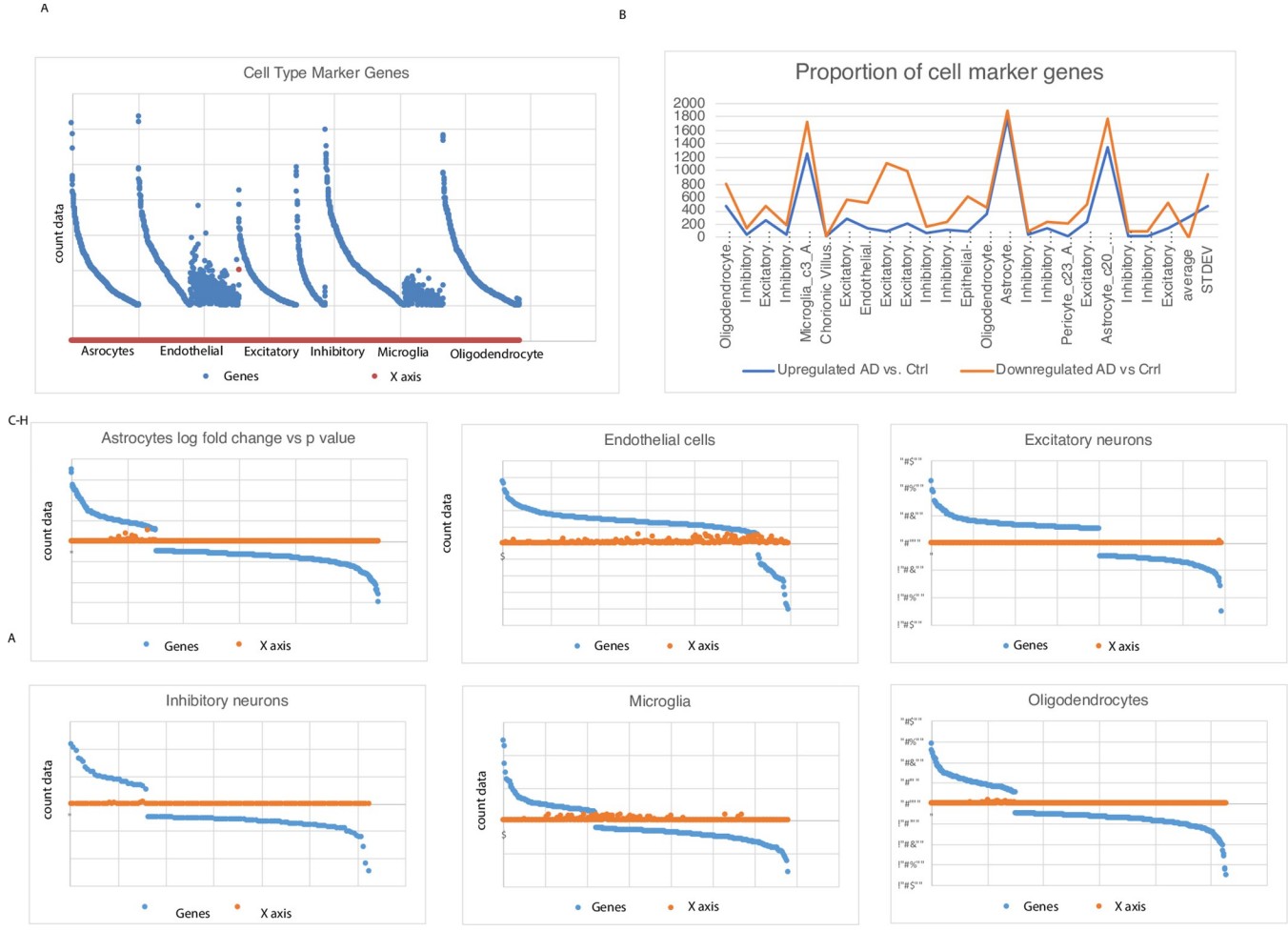

**Fig 6. Cell type specific marker genes.** Left panel: Plot of cell type marker genes detected as altered in AD compared to control samples based on the single cell Alzheimer's seq data analysis (of 7 neuronal cell type gene markers). Top right panel: Proportion (percent) of astrocytes and oligonucleotide cell specific marker genes. X axis: Cell types, y axis: Seq count data. The expression data is plotted in blue. Right: Another plot of the promotion of cell marker genes (blue/; upregulated in AD vs control, red: Down) Y axis range: 0–2000 X axis: Different cell-type marker gene groups (e.g oligodendrocytes, inhibitory excitatory neurons microglia astrocytes). Bottom left: Count data gene plots (fold change vs p value). Astrocyte cell marker genes fold change vs p value, right panel: Endothelial cell marker genes. Right panel: Excitatory neurons. Lower panel: Inhibitory neurons, microglia and oligodendrocyte marker genes. Blue: Alzheimer's brain samples, orange: Control. Y axis: Gene count values of cell type marker genes, x axis: 6 cell type marker gene categories (left—astrocytes, endothelial cell markers, excitatory neurons, inhibitory neurons microglia and oligodendrocytes).), astrocyte and A) A gene network of macroglia and oligodendrocytes. The genes are plotted in blue and red and gene symbols are given. The genes are marked in blue or red according to the sample type (control/ Alzheimer's). Down panel plots: 7 cell types count data.

*SOX2* mediated transcriptional networks in astroglia. In addition reinforcing prior Alzheimer's disease findings, our current results nominate new networks mediating disease within discrete cell populations.

In the future, imaging of post-mortem Alzheimer's patients may verify brain cellular morphology/cell volume of vascular changes in the Alzheimer's compared to control samples. [11]. Additionally, Alzheimer's mice model (e.g *VAcht* Knock-out mice) RNA-Seq may serve as a comparative dataset [12].

Our current findings also provide insights into the cellular signalling in Alzheimer's disease. The results we produced in our current study are a valuable resource to the scientific community We have classified genes such as *ST18*, *VWF GFAP* and *SYNRP*. We have found differential expression changes in Pericyte neurons, astrocyte, microglia and dendrocytes, including in

the gene *CXCL1* and synaptic pathways. Efferocytosis is the process in which apoptotic cells are removed by phagocytic cells. Efferocytosis can be performed not only by 'professional' phagocytic cells such as macrophages or dendritic cells, but also by many other cell types including epithelial cells and fibroblasts. To distinguish them from living cells, apoptotic cells carry specific cellular signals, such as the presence of phosphatidyl serine. Efferocytosis triggers specific downstream intracellular signal transduction pathways, for example resulting in anti-inflammatory, anti-protease and growth-promoting effects. Conversely, impaired efferocytosis has been linked to autoimmune disease and tissue damage. Efferocytosis results in production by the ingesting cell of mediators such as hepatocyte- and vascular endothelial growth factor, which are thought to promote replacement of the dead cells. Efferocytosis triggers specific downstream intracellular signal transduction pathways, for example resulting in anti-inflammatory, anti-protease and growth-promoting effects. Conversely, impaired efferocytosis has been linked to autoimmune disease and tissue damage.

Microglia cells are well known as involved in Alzheimer disease (AD) [13]. However in our current study we also report involvement of other cell types including astrocytes, oligodendrocytes and neuronal cell type marker genes. Previously, Defective efferocytosis has been demonstrated in neurological diseases as cystic fibrosis and bronchiectasis, chronic obstructive pulmonary disease, asthma and idiopathic pulmonary fibrosis, rheumatoid arthritis, systemic lupus erythematosus, glomerulonephritis and atherosclerosis. UMAP cluster analysis revealed that the proportions of astrocytes, excitatory neurons, inhibitory neurons, microglia, and oligodendrocytes were similar between the Alzheimer's and neurological controls brain samples. Thus, perhaps this process is also involved in AD, as well as in inflammation. Future experiments might extend the cohort size by comparing the results from our study to other larger case/cohort Alzheimer's single cell studies. (e.g [10,14,15]).

## Conclusions

In our current manuscript we perform single-cell RNA-Seq from frontal cortex of from two individuals with AD and two controls. This is a pivotal study with a small dataset but potentially valuable. The major finding is that cell-specific markers of glial cells are elevated in Alzheimer's samples compared to controls which is consistent with previous literature. The single time point and limited sample size mean that it is difficult to infer which observed changes are upstream in disease pathogenesis and which are the downstream consequence of neurodegeneration per-se. We have also discovered an increase in the frequency of microglia, astrocytes and oligodendrocytes which has been previously reported in AD including in genome-wide transcriptome studies. However we currently cannot assess whether the observed changes are downstream or upstream the disease progression. This is a potential caveat of our current research. Regarding the glial cell changes, our main conclusion is that the glial cells may be reactive in the disease and not causal.

## Supporting information

**S1 Graphical abstract.**
(PDF)

**S1 Table. ST3 –Samples clinical information/description table (short version/0).**
(XLSX)

**S2 Table. ST4 –AD and control samples clinical info (full version) the study selected samples are marked in blue.**
(XLSX)

**S1 File. Ethics approval–Title—Material Transfer Agreement for the Supply of Human cells, legend—MTA document.**
(PDF)

**S2 File. Ethical recommendation letter–Title—Confirmation of ethical opinion legend–samples ethical recommendation, source—the university of Edenborough/ Edenborough brain bank.** IRAS project ID 199294.
(PDF)

## Acknowledgments

We thank Dr. Nathan Salomonis for his assistance with the use of AltAnalyze the single-cell RNA-Seq analyses. We thank the UK brain expression consortium (UK brain bank) for the control brain samples and the Edinburgh brain bank for the collaboration and Alzheimer's post-mortem brain samples. Sequencing was carried out at the UCL Genomics Core. We also thank Dr. Carlo Sala Frigerio (CSF) (UCL ION, London UK) for arranging the Chromium single cell Seq experiment at the UCL Genomics Centre. We also acknowledge Mr. Weis, Michael (Chad), Cincinnati USA for deposition of the fastq files to the GEO database.

## Author Contributions

**Conceptualization:** Lilach Soreq, John Hardy.

**Data curation:** Lilach Soreq.

**Formal analysis:** Lilach Soreq.

**Funding acquisition:** John Hardy.

**Investigation:** Lilach Soreq, Hannah Bird, Wael Mohamed.

**Methodology:** Lilach Soreq.

**Resources:** Lilach Soreq.

**Software:** Lilach Soreq.

**Supervision:** Wael Mohamed.

**Visualization:** Lilach Soreq.

**Writing – original draft:** Lilach Soreq.

**Writing – review & editing:** Lilach Soreq, Hannah Bird, Wael Mohamed, John Hardy.

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
