## [Decision Letter · Decision Letter 0]

3 Aug 2022

PONE-D-22-19070Single cell RNA sequencing analysis of human Alzheimer’s disease brain samples reveal neuronal and glial specific cells differential expressionPLOS ONE

Dear Dr. Soreq,

Thank you for submitting your manuscript to PLOS ONE. After careful consideration, we feel that it has merit but does not fully meet PLOS ONE’s publication criteria as it currently stands. Therefore, we invite you to submit a revised version of the manuscript that addresses the points raised during the review process.

We look forward to receiving your revised manuscript.

Kind regards,

Jinhui Liu

Academic Editor

PLOS ONE

Journal Requirements:

"none"

Reviewers' comments:

Reviewer's Responses to Questions

**Comments to the Author**

1. Is the manuscript technically sound, and do the data support the conclusions?

Reviewer #1: Partly

Reviewer #2: Yes

2. Has the statistical analysis been performed appropriately and rigorously? 

Reviewer #1: Yes

Reviewer #2: Yes

3. Have the authors made all data underlying the findings in their manuscript fully available?

Reviewer #1: Yes

Reviewer #2: Yes

4. Is the manuscript presented in an intelligible fashion and written in standard English?

Reviewer #1: Yes

Reviewer #2: Yes

5. Review Comments to the Author

Reviewer #1: The manuscript by Soreq et al. describes single cell RNA sequencing results of two human AD brain samples compared to two age matched control brain samples. While the analysis and the data can be a valuable resource for the scientific community there are major inconsistencies with respect to the samples used for the analysis, typographical error and some additional points that should be addressed by the authors.

1. In the abstract the authors state that two AD cases at Braak stage I and II and age-matched controls were used. In the methods section however the authors say that the AD samples were at Braak stages III and IV. The authors need to clarify this. Furthermore, the reported staging for the AD cases is incomplete. CERAD score and Thal phase also need to be reported (also for the control cases). What was the APOE status of the cases?

2. The authors say that they profiled three brain regions per brain, but focused the analysis only on the frontal cortex. This may be a good approach to streamline data presentation, but at a certain point the available data for the other brain regions should be included in the analysis and regional variation should be analyzed. The authors compare their data to available datasets from mouse and human, so a comparison of brain region specific data should also be included.

3. The authors state that "microglia were consistently decreased between the two patients". Looking at the cell cluster frequency graph in Figure 2, it appears that the frequency of microglia is highly variable between the samples, but overall AD cases seem to have a higher frequency of microglia clusters (which is expected). What does the above mentioned sentence refer to? This should be clarified.

4. According to the methods section both AD cases were male, while the control cases were male and female. Although the study is not powered to detect gender specific differences this should at least be commented on, as more and more evidence emerges that sex differences with respect to AD related changes are profound.

5. Throughout the manuscript there are many incomplete or duplicated sentences as well as typos. For example on page 4: "...response in human vs mice MG depleted mice..." or on page 12, where the whole sentence referring to the value of the resource for the scientific community appears to be duplicated. This hampers the overall readability of the manuscript.

Reviewer #2: In this study, the authors identify transcriptional networks impacted into distinct neuronal populations in human Alzheimer’s disease and control brains. Although the number of brains is small, the strength is that the authors surveyed gene expression differences in over 25,000 single-nuclei collected from the brains of disease patients in Braak stage I and II hippocampal brain samples. This study is timely and will be of interest to a large readership involved in neurodegenerative disease research and, more largely, in molecular neurosciences.

Major concern:

A major challenge is to understand the functional significance of gene expression differences observed in human neurodegenerative disease conditions. Although some of these differences may be disease specific, some others may be shared across disease conditions, particularly if they correspond to homeostatic responses. Here, the comparison of mouse and human data is needed. To address the gap between gene expression data and functional data, the authors may want to compare their data with functional data obtained in mouse and invertebrate models of neurodegenerative disease insults as, should overlaps be detected, this might add much insight into molecular pathogenesis, which will be of interest to a broader readership, e.g. researchers involved in AI, big data and systems-modelling. Below are papers that are relevant to that end as they rely on cell-type specific data in invertebrate and mouse models and as they focus on compensatory and homeostatic genes and pathways that may be commonly engaged in response to several types of neurodegenerative disease triggers.

https://pubmed.ncbi.nlm.nih.gov/29936182/

https://pubmed.ncbi.nlm.nih.gov/33618800/

6. PLOS authors have the option to publish the peer review history of their article (what does this mean?). If published, this will include your full peer review and any attached files.

Reviewer #1: No

Reviewer #2: No

---

## [Author Response · Author response to Decision Letter 0]

31 Aug 2022

04/08/22

Re-battle letter (PLoS ONE) Soreq L et al 

Paper title Single cell RNA sequencing analysis of human Alzheimer’s disease brain samples reveal neuronal and glial specific cells differential expression

Reviewer 1

Comment: The manuscript must describe a technically sound piece of scientific research with data that supports the conclusions. Experiments must have been conducted rigorously, with appropriate controls, replication, and sample sizes. The conclusions must be drawn appropriately based on the data presented.

Partly 

Reply: Yes, the paper is technically sound the RNA was extracted from the post mortem samples in high quality and the Chromium Seq experiment was done in high standard. The experimental data supports the conclusions. The manuscript describes a technically sound piece of scientific research with data that supports the conclusions. Experiments have been conducted rigorously, with appropriate controls, replication, and sample sizes (n=4 in this case). The conclusions are drawn appropriately based on the data presented.

Comment: The authors compare their data to available datasets from mouse and human, so a comparison of brain region specific data should also be included.

Reply: Indeed this was the case, in fact comparison also to cell type specific mouse brain RNASeq dataset was done (brain-map.org) and in fact we are not aware of any other available brain region specific seq publicly available data as for the time of paper submission (human brain RNA expression data and in fact mice as well is still quiet rare, in particular large datasets). But we did analyze we have analysed the GEO dataset GSE157827 (composed of a total of 21 samples, 13 Alzheimer’s and 9 neurological controls (figure 6).

Comment: According to the methods section both AD cases were male, while the control cases were male and female. Although the study is not powered to detect gender specific differences this should at least be commented on, as more and more evidence emerges that sex differences with respect to AD related changes are profound.

Reply: Apologize both AD samples were males (marked in blue in ST1), indeed we are aware of gender related expression differences in RNA expression values this is why we included only males in the current study. 

5. Throughout the manuscript there are many incomplete or duplicated sentences as well as typos. For example on page 4: "...response in human vs mice MG depleted mice..." or on page 12, where the whole sentence referring to the value of the resource for the scientific community appears to be duplicated. This hampers the overall readability of the manuscript.

Reply: Thank you for this comment. We went over the paper text thoroughly removed duplicates and corrected spelling/grammar were needed. 

Reviewer #2: In this study, the authors identify transcriptional networks impacted into distinct neuronal populations in human Alzheimer’s disease and control brains. Although the number of brains is small, the strength is that the authors surveyed gene expression differences in over 25,000 single-nuclei collected from the brains of disease patients in Braak stage I and II hippocampal brain samples. This study is timely and will be of interest to a large readership involved in neurodegenerative disease research and, more largely, in molecular neurosciences.

Reply: Thank you.

Major concern:

A major challenge is to understand the functional significance of gene expression differences observed in human neurodegenerative disease conditions. Although some of these differences may be disease specific, some others may be shared across disease conditions, particularly if they correspond to homeostatic responses. Here, the comparison of mouse and human data is needed. To address the gap between gene expression data and functional data, the authors may want to compare their data with functional data obtained in mouse and invertebrate models of neurodegenerative disease insults as, should overlaps be detected, this might add much insight into molecular pathogenesis, which will be of interest to a broader readership, e.g. researchers involved in AI, big data and systems-modelling. Below are papers that are relevant to that end as they rely on cell-type specific data in invertebrate and mouse models and as they focus on compensatory and homeostatic genes and pathways that may be commonly engaged in response to several types of neurodegenerative disease triggers.

Reply: 

Thank you we appreciate this important note. In the current study we compared our dataset to 3 external datasets. However we feel that further comparison to mouse nd invertebrate models of neurodegenerative diseases would take out the focus from the paper. We did apply advanced bioinformatic statistical and computational data analyses on our data so our paper may be of interest to AI big data and system modelling researchers as is.

General comment: we formatted the references to PLoS ONE style and included the supporting table info after the acknowledgments, the funding and competing interests sections are included separately. We changed heading text size to 14. 

We also

---

## [Decision Letter · Decision Letter 1]

6 Oct 2022

PONE-D-22-19070R1Single-cell RNA sequencing analysis of human Alzheimer’s disease brain samples reveal neuronal and glial specific cells differential expressionPLOS ONE

Dear Dr. Soreq,

Thank you for submitting your manuscript to PLOS ONE. After careful consideration, we feel that it has merit but does not fully meet PLOS ONE’s publication criteria as it currently stands. Therefore, we invite you to submit a revised version of the manuscript that addresses the points raised during the review process.

We look forward to receiving your revised manuscript.

Kind regards,

Jinhui Liu

Academic Editor

PLOS ONE

Additional Editor Comments:

Based on the comments of reviewers, it is recommended major revision.

Reviewers' comments:

Reviewer's Responses to Questions

**Comments to the Author**

1. If the authors have adequately addressed your comments raised in a previous round of review and you feel that this manuscript is now acceptable for publication, you may indicate that here to bypass the “Comments to the Author” section, enter your conflict of interest statement in the “Confidential to Editor” section, and submit your "Accept" recommendation.

Reviewer #1: (No Response)

Reviewer #2: All comments have been addressed

2. Is the manuscript technically sound, and do the data support the conclusions?

Reviewer #1: Yes

Reviewer #2: Yes

3. Has the statistical analysis been performed appropriately and rigorously? 

Reviewer #1: Yes

Reviewer #2: Yes

4. Have the authors made all data underlying the findings in their manuscript fully available?

Reviewer #1: Yes

Reviewer #2: Yes

5. Is the manuscript presented in an intelligible fashion and written in standard English?

Reviewer #1: Yes

Reviewer #2: Yes

6. Review Comments to the Author

Reviewer #1: The authors have only addressed minor points, but did not provide adequate answers to major concerns raised in comments 1-3. These comments with respect to staging of AD pathological changes in the patient samples and other aspects of the data (why were only data for the frontal cortex shown? What does the sentence about decreased microglia in patients refer to?) are crucial to assess the impact of the study and should be answered in detail.

1. In the abstract the authors state that two AD cases at Braak stage I and II and age-matched controls were used. In the methods section however the authors say that the AD samples were at Braak stages III and IV. The authors need to clarify this. Furthermore, the reported staging for the AD cases is incomplete. CERAD score and Thal phase also need to be reported (also for the control cases). What was the APOE status of the cases?

2. The authors say that they profiled three brain regions per brain, but focused the analysis only on the frontal cortex. This may be a good approach to streamline data presentation, but at a certain point the available data for the other brain regions should be included in the analysis and regional variation should be analyzed. The authors compare their data to available datasets from mouse and human, so a comparison of brain region specific data should also be included.

3. The authors state that "microglia were consistently decreased between the two patients". Looking at the cell cluster frequency graph in Figure 2, it appears that the frequency of microglia is highly variable between the samples, but overall AD cases seem to have a higher frequency of microglia clusters (which is expected). What does the above mentioned sentence refer to? This should be clarified.

Reviewer #2: The authors have addressed my concerns and the revised paper may be considered for publication in the journal

7. PLOS authors have the option to publish the peer review history of their article (what does this mean?). If published, this will include your full peer review and any attached files.

Reviewer #1: No

Reviewer #2: No

---

## [Author Response · Author response to Decision Letter 1]

6 Oct 2022

i prepared a rebuttal letter and corrected the paper. Thank you

---

## [Decision Letter · Decision Letter 2]

1 Nov 2022

Single-cell RNA sequencing analysis of human Alzheimer’s disease brain samples reveals neuronal and glial specific cells differential expression

PONE-D-22-19070R2

Dear Dr. Soreq,

We’re pleased to inform you that your manuscript has been judged scientifically suitable for publication and will be formally accepted for publication once it meets all outstanding technical requirements.

Kind regards,

Jinhui Liu

Academic Editor

PLOS ONE

Additional Editor Comments (optional):

Reviewers' comments:

Reviewer's Responses to Questions

**Comments to the Author**

1. If the authors have adequately addressed your comments raised in a previous round of review and you feel that this manuscript is now acceptable for publication, you may indicate that here to bypass the “Comments to the Author” section, enter your conflict of interest statement in the “Confidential to Editor” section, and submit your "Accept" recommendation.

Reviewer #1: All comments have been addressed

2. Is the manuscript technically sound, and do the data support the conclusions?

Reviewer #1: Yes

3. Has the statistical analysis been performed appropriately and rigorously? 

Reviewer #1: Yes

4. Have the authors made all data underlying the findings in their manuscript fully available?

Reviewer #1: Yes

5. Is the manuscript presented in an intelligible fashion and written in standard English?

Reviewer #1: Yes

6. Review Comments to the Author

Reviewer #1: The authors have adressed all my comments and updated the manuscript accordingly. Some important data pertaining to neuropathological characterization of the cases were not available.

7. PLOS authors have the option to publish the peer review history of their article (what does this mean?). If published, this will include your full peer review and any attached files.

Reviewer #1: No

---

## [Editor Report · Acceptance letter]

23 Nov 2022

PONE-D-22-19070R2 

Single-cell RNA sequencing analysis of human Alzheimer’s disease brain samples reveals neuronal and glial specific cells differential expression 

Dear Dr. Soreq:

I'm pleased to inform you that your manuscript has been deemed suitable for publication in PLOS ONE. Congratulations! Your manuscript is now with our production department. 

Kind regards, 

on behalf of

Dr. Jinhui Liu 

Academic Editor

PLOS ONE